# Effect of the Solution Temperature on the Precipitates and Grain Evolution of IN718 Fabricated by Laser Additive Manufacturing

**DOI:** 10.3390/ma13020340

**Published:** 2020-01-11

**Authors:** Yu Cao, Pucun Bai, Fei Liu, Xiaohu Hou, Yuhao Guo

**Affiliations:** 1College of Materials Science and Engineering, Inner Mongolia University of Technology, Hohhot 010051, China; ngdcaoyu@126.com (Y.C.); ngdliufei@163.com (F.L.); houxiaohuhu@163.com (X.H.); gyh512480516@163.com (Y.G.); 2Department of Mechanical and Electrical Engineering, Hulunbiur College, Hulunbiur 021000, China; 3Engineering Research Center for the Safe Exploitation and Comprehensive Utilization of Mineral Resources at Universities of Inner Mongolia Autonomous Region, Hulunbiur 021000, China

**Keywords:** SLM, IN718, precipitates, grain feature, recrystallization

## Abstract

The effects of the solution heat treatment temperature on the precipitates, grain boundary evolution and response of the microhardness of Inconel 718 (IN718) superalloy fabricated by selective laser melting (SLM) were investigated. It was found that: (1) The long-chained Laves phases formed in the as-deposited condition dissolved into the matrix when the solution temperature rises above 980 °C. (2) The width-to-length ratio was maintained at approximately 1.6 when the solution was heated from 980 °C to 1080 °C, and dropped down to 1.03 when heated to 1130 °C. (3) Low-angle grain boundaries kept the same number fraction of 65% from 980 to 1080 °C as the as-deposited condition, and decreased dramatically from 1090 to 1130 °C to 4%. (4) Annealing twin boundaries occurred at 1090 °C with a number fraction of 3%, and quickly increased to 65% when heated to 1130 °C. It is concluded that the static recrystallization of IN718 fabricated by selective laser melting (SLM) occurred at 1090 °C and fast proceeded to full recrystallization at 1130 °C. The forming of annealing twins accompanies the recrystallization process and is an effective way to refine the recrystallized grain size.

## 1. Introduction

Inconel 718 (IN718) superalloy is mainly strengthened by precipitate of coherent, ordered metastable γ″ phase (Ni_3_Nb) with a body-centered tetragonal (bct) DO_22_ structure [1,2,3]. It has been widely used for hot section rotating components of gas turbines due to its excellent corrosion-resistant, thermal and strength properties under extreme working conditions. Selective laser melting is a promising additive manufacturing (AM) technique which can form metal parts with a complex shape and hollow structure directly [4,5,6]. The fast solidification rate of the selective laser melting (SLM) process can result in much finer microstructure and uniform element distribution. Thus, the macro segregation can be avoided, but micro segregation still exists during this process. Aiming to optimize the microstructure generated from SLM process, lots of researchers have investigated the heat treatment regime of SLMed parts [7,8,9,10]. The heat treatment regime of solution plus aging, which is similar with the forging IN718 parts, has been accepted by most researchers. However, most concerns are focused on the dissolution of detrimental precipitate generated from the SLM process and the precipitation of the strengthening phase γ″ during the aging process, and little attention has been paid to the grain boundary evolution during the solution heat treatment process. In fact, during the high temperature solution process, along with element homogenization, the grain boundary evolution is inevitable, which will also affect the mechanical properties of the parts. The presence of low-angle grain boundaries can produce strain strengthening, which will enhance the microhardness of the materials. In addition, Pande et al. [11] confirmed on the basis of experiments that annealing twin boundaries at room temperature can reduce the effective grain size and increase the room temperature yield strength of the alloy. Yuan et al. [12,13] found in a new type of Ni-Co based deformed superalloys that during the creep process of alloys, the annealed twin boundaries can hinder dislocation slippage and improve the high temperature creep life of alloy. Thus, investigation on the grain boundary evolution of this process is very meaningful. This research aims to explore the effect of the solution heat treatment temperature on both the precipitates and grain boundary migration and the response of the sample hardness, revealing the combined influence of precipitates and grain boundaries on the mechanical properties.

## 2. Materials and Methods

### 2.1. SLM Experiment

The chemical composition of gas atomized IN718 powder is revealed in Table 1. The SLM experiment was conducted on the EOS M280 rapid forming machine (EOS Gmbh, Munich, German). The material of substrate was 45 steel with dimensions of 250 mm × 250 mm × 30 mm. In order to avoid deformation of the sample, the substrate was preheated to 80 °C. The parameters of SLM process were as follows: 0.2 mm hatch spacing, 0.04 mm layer thickness, 350 W laser power, and 1000 mm/s laser scanning speed, stripped scanning strategy shown in Figure 1a, with 67° turning angles layer by layer which have already been explored in our previous research [14]. As shown in Figure 1, the rectangular sample with dimensions of 10 mm × 10 mm × 100 mm was cut into 20 pieces to do the following heat treatment and microstructural observation.

### 2.2. Heat Treatment

Different solution heat treatments parameters are illustrated in Table 2. Solution heat treatments were conducted between 930 °C and 1230 °C for 1h, after which the samples were quenched in the water to room temperature (WC). All the solution heat treatment procedures were performed in a KSL-1400X furnace (MTI Corporation, Shenyang, China).

### 2.3. Microstructural Investigation

Scanning electron microscopy (SEM) (Quanta 650, FEI, Hillsboro, OR, USA) was used to observed the microstructure and precipitate distribution, operating at 20 kV and equipped with a secondary electron signal for imaging. For electron backscatter diffraction (EBSD) observation, the specimens were mechanically polished, followed by electropolishing in a solution of 10% perchloric acid and 90% ethanol at 25 V for 30 s to produce a strain-free surface. EBSD measurement was performed with the following parameters: a spot size of 6.0, accelerating voltage 20 kV, and step size of 1.2 μm. Microstructure analyses were carried out using a fully automated EBSD system with KHL channel 5 software (5.0, Oxford Instruments, Abingdon, Oxfordshire, UK). The misorientation angles (MAs) less than 2° were deleted to eliminate the false boundaries caused by the orientation noise. The MAs between 2° and 10° were defined as low-angle boundaries (LABs), and the MAs above 10° defined as high-angle boundaries (HABs). The kernel average misorientation angles (KAMAs) below 1° (low KAMAs) denoted the low local misorientation versus the high ones, compared with KAMAs above 1° (high KAMAs). Thin foils with a diameter of 3 mm for transmission electron microscopy (TEM) were prepared by mechanically grinding down to 50 μm and further electropolishing down to electron transparency. The TEM characterization was performed using a TEM (Talos 200, FEI, Hillsboro, OR, USA), operating at an accelerating voltage of 200 kV to make the crystal structure determination.

### 2.4. Microhardness Test

The hardness measurement of the samples of different conditions was performed by a hardness testing device (HVS-30, Shanghai Gaozhi Precision Instrument Co., Ltd., Shanghai, China) under a load of 0.98 N for a dwell time of 15 s with an interval of 0.1 mm.

## 3. Results and Discussion

### 3.1. Precipitates

During the SLM process, micro segregation is easy to form because of the high solidification rate and small laser spot size of SLM process [15]. Niobium (Nb) is easy to be trapped in the liquid phase during the solidification process due to the lower value of the partition coefficient [16]. Thus, Laves phases enriched of Nb have always been observed in the as-deposited IN718. Laves phases have been proved to be detrimental to the mechanical properties of SLMed IN718 by many researchers [17,18]. In addition, Laves phase has exhausted much Nb which is used to form the strengthening phase γ″ (Ni_3_Nb) in the following aging treatment process. Trying to release Nb into the matrix during the solution heat treatment process is essential to improve the property of SLMed IN718.

Figure 2 shows the precipitate distributions of IN718 samples in the as-deposited condition and different solution heat treatment conditions. In Figure 2a, the Laves phase distributed along the grain boundaries and subgrain boundaries to form a chained-like connection, which has clearly displayed the grain boundaries and dendrite-cellular substructure inside of the grains. The forming of the finer substructures was attributed to the high cooling rate. Figure 2b illustrates more precipitates with profuse morphologies. The acicular precipitates were coarse γ″ and the short-rod precipitate was δ (Ni_3_Nb), which is in accordance to the references [19,20]. Under the solution heat treatment condition of 930 °C/h, δ phase precipitates from the matrix, since the precipitation temperature of δ is between 750 °C and 950 °C in IN718. As the equilibrium phase of γ″, δ phase has proven to be detrimental to the high-temperature property of IN718. However, moderate δ could inhabit the migration of the grain boundaries to maintain the grain size [21]. When the solution temperature increased to 980 °C, most of the Laves phases have dissolved into the matrix and regular arranged short-rod δ was observed along the grain boundaries. From 1030 °C to 1230 °C, the distribution of precipitates showed almost the same condition. A small amount of undissolved particle Laves phases were found along the subgrain boundaries and undissolved bulk Laves phases were found along the grain boundaries. This is mainly because in the as-deposited condition, the grain boundaries accumulated more Laves phases than the subgrain boundaries with bigger dimension and higher concentration of Nb. During this temperature interval from 1030 °C to 1230 °C, with the rising of the temperature, the morphology of the Laves phase distribution is basically unchanged. Even the higher solution temperature could not dissolve all the Laves phases. To be noted, incipient melting phenomenon has been found in the 1280 °C solution condition, as indicated in Figure 2h. According to the reference [22,23,24], eutectic reaction of Laves phase occurs at 1160–1200 °C and the solidus temperature of Laves phase is 1255 °C. In fact, 1180 °C is the critical temperature to occur the incipient melting of Laves phases in the wrought IN718. However, the incipient melting did not occur until 1230 °C in the SLMed IN718. During the SLM process, the solidification rate is so high that the demission of the Laves phase is much smaller than in the conventional forging process, which means it is easier to dissolve the Laves phase generated from the SLM process than in the conventional forging process. Thus, the incipient melting occurs above the solidus temperature of Laves phase in the SLM process.

### 3.2. Grain Features

Though the solution heat treatment can homogenize the element distribution of the sample, the static recrystallization phenomenon triggered by the solution heat treatment also can affect the microstructure and the mechanical properties of the samples. During the SLM process, each layer has experienced heating and cooling repeatedly, resulting in the residual stress in the samples [25]. The residual stress is employed as the driving force in the following solution heat treatment process to trigger the static recrystallization at the appropriate temperature and time [26]. Different from conventional forging and extruding processes, the samples fabricated by the SLM process haven’t experienced the big plastic deformation, and the deformation energy storage is relatively low. The grain size and grain distribution have special features after static recrystallization.

#### 3.2.1. Grain Size

Figure 3 is the inverse-pole figure map of the samples in the as-deposited and different solution heat treatment conditions. A color-coded triangle is displayed in the top right corner. In Figure 3a, the microstructure of the as-deposited sample is composed of a significant number of elongated columnar grains and small quantity of equiaxed grains. Since most of the grains are columnar grains, the average grain width along X axis direction and average grain length along Z axis direction were measured to characterize the grain size features by the linear intercept method. The average grain width was about 9.09 μm and grain length was 15.11 μm, and the aspect ratio (length to width ration) was about 1.66 of the as-deposited condition. In the SLM process, the heat dissipation direction is almost along the building direction, so most of the grains epitaxially grow along same direction with little tilting angles to form the columnar grains. The maximum length of columnar reached approximately 200 μm, which means the grain was epitaxially grown across 6–7 layers. In Figure 3b–d, it is difficult to distinguish the recrystallization nucleus and initial small equiaxed grains of as-deposited condition. It is ambiguous whether the recrystallization process occurred. From 1080 °C to 1130 °C, the 50 °C temperature interval proceeded the recrystallization process quickly. In order to see the detailed transformation, the results of grain evolution of heat solution treatment conditions of 1080 °C/h to 1130 °C/h with 10 °C intervals are illustrated in Figure 4. Figure 5 reveals the aspect ratio of the samples in different solution treatment conditions. From 980–1080 °C, the aspect ratio showed no obvious change, holding on around 1.6. Under the solution heat treatment condition of 1130 °C/h in Figure 3e, the grain width became bigger and grain morphology changed into equiaxed grain, while the aspect ratio decreased to 1.03, which means complete recrystallization finished. To be noted, some annealing twins in the interior of the grains have also been observed under this condition.

Figure 4a shows almost the same condition with Figure 3d; the mix grains occupied the whole section. Even some recrystallization nucleuses accumulated along the grain boundaries of the columnar grains, but some un-recrystallized coarse columnar grains with some substructures in them were dominant of this condition. When heated to 1110 °C, distinct changes occurred: fully recrystallized grains with annealing twins inside and un-recrystallized columnar grains with substructures coexisted in the grain morphology. Due to the uneven grain distribution in the as-deposited condition, the stored energy which is the driving force of recrystallization process was not homogeneous. Thus, the extent of recrystallization was different in different areas. When heated to 1120 °C, more recrystallized grains and less deformed grains showed on this section. The full recrystallized grains with annealing twins became the only grain morphology at 1130 °C. The annealing twins only existed in the unstrained recrystallized grain, which means the recrystallization process is accompanied by the formation of twins. It is a vital way to refine the grain size during the recrystallization process.

The deformation amount can affect the recrystallization temperature is one aspect. The bigger the deformation is, the more energy storage, and the bigger the driving force of the recrystallization process. Thus, a lower recrystallization temperature is needed. However, during the SLM process, deformation is much smaller, which leads to higher recrystallization temperature. The visible recrystallization process occurred in 1090 °C. In addition, the precipitates concentrated along the grain boundaries and subgrain boundaries can also inhabit the grain migration and the movement of the dislocation. During 980–1080 °C, even most of the Laves phases were dissolved into the matrix. There were still some undissolved small particles distributed along the grain boundaries and subgrain boundaries, which inhabited the migration of the grain boundaries. As a result, the recrystallization temperature is raised. Finally, the initial grain sizes of the as-deposited condition were big compared to the traditional forging process, and the total area per unit volume was small. Thus, less recrystallization nucleation sites were available.

#### 3.2.2. Misorientation Angle Evolution 

Figure 3 also shows the distribution of the LABs and HABs, which are represented by the gray full lines and black full lines, respectively.

Figure 3a indicates the columnar grains with relatively high density of LABs, which is in accordance with microstructure in Figure 2a. During the solidification process of SLM, the substructure of the grain is composed of columnar dendrites and cellular dendrites which are divided by the LABs. From 980 °C to 1080 °C, the LABs in the columnar grains maintained the same status. When heated to 1130 °C, the LABs decreased significantly and most of HABs were twin boundaries. Figure 6 shows the number fraction (NF) of LABs, HABs and annealing twins’ boundaries of different heat treatment statuses. The NF of the LABs in as-deposited condition and 980 °C/1030 °C/1080 °C fluctuated around 34% with small deviation. After 1090 °C, the LABs began to decrease while the NF of the HABs increased. The NF of annealing twins showed the same trends with the HABs. After 1120 °C, the NF of annealing twins was stable at 60%. This result indicates that the recovery process didn’t occur during 980–1080 °C, and after 1090 °C the recrystallization process emerged.

IN718 is a kind of alloy with medium or low stacking fault energy. For metals with low and medium layer fault energy, the softening mechanism is mainly recrystallization, not recovery [27]. When the stacking fault energy is low, the total dislocation can be easily decomposed into an extended dislocation with a large stack fault width. The recovery process mainly relies on dislocations to offset the dislocations of each other after climbing, slipping or cross-slipping. The process of reducing the dislocation density is manifested by the reduction of LABs. It is very hard for the extended dislocations to climb, slip or cross-slip, because the extended dislocations contain staking faults. If so, the reverse process of the total dislocation decomposition called bundle is needed. However, the lower the stacking fault energy, the wider the extend dislocation, and the more difficult the bundle. Thus, it is difficult to recover.

Figure 7 gives the KAMA map of samples of different solution treatment conditions to estimate the plastic strain. The color-coded bars are shown in the top left corner of Figure 7a. KAMAs reveal the local variation of lattice orientation in a given area defined by the investigators, and it is a good indicator of plastic strains in crystals [28,29,30,31]. Figure 7 suggests that the higher value of KAMAs mainly focused on the columnar grains with lots of LABs in them. As the arrangements of dislocations, the LABs can be roughly associated with the plastic strains. Figure 7b shows the number fraction (NF) of the KAMAs on the as-deposited condition. The low KAMAs donated 56%. This existing internal stress can motivate recrystallization nucleation when proper solution temperature is provided. Figure 7a–e indicates that with the increase of solution temperature, the distribution change of KAMAs was not obvious; this result is quantitatively counted in Figure 8. The NF of low KAMAs of heat-treated conditions was little higher than that of the as-deposited condition, but was still less than 0.65. Figure 7f shows the uniform distribution of the KAMAs until heated to 1130 °C, since the sample was totally recrystallized under this condition. After recrystallization, the unstrained equiaxed grains replaced the columnar grains. The residual stress was totally released. To make the detailed description of the recrystallization process, the KAMAs distribution from 1090–1120 °C is given in Figure 9. Figure 10 shows the NF of the low KAMAs from 1090 ℃ to 1120 ℃. The NF of the low KAMAs increased from 0.52 to 0.92, indicating the residual stress gradually decreased. 

#### 3.2.3. Annealing Twins

Figure 9a shows the lamella-like straight <111> 60°annealing twins in the recrystallized grains. The annealing twin boundary is represented by red full lines. Some annealing twins nucleated on the trigeminal grain boundaries, shown by the white arrows. With the growth of the recrystallized grains, the annealing twin extended to the grain boundary. From 980 °C to 1080 °C, the NF of the annealing twin boundary held on 1%, which means the forming of annealing twins is accompanied with grain growth process of recrystallization instead of recovery. With the increasing of solution temperature from 1090 °C to 1130 °C, the NF of annealing twins boundary dramatically increased from 3% to 65% as illustrated in Figure 6. On the one hand, the rising of temperature can increase the amount of recrystallization grains. On the other hand, the higher the temperature is, the bigger the grain size is. Bigger grain size can enhance the amount of annealing twins. When heated to 1110 °C, another annealing morphology appeared. The isolate-island-like annealing twins indicated by the white arrows in Figure 9c were isolated inside of the grain, not connecting to the grain boundary. Figure 9d demonstrates the annealing twin distribution of 1130 °C. Most of the annealing twins connected with grain boundaries, while few step-like twins appeared under this temperature. The annealing twin boundaries became more and longer. Thus, the annealing twin boundaries connected with each other or with the grain boundaries to form a seal grain, which has refined the grain size of the recrystallized grains. The twins that were mainly present in the forged IN718 alloy were deformed twins. A large number of deformed twins were observed in the initial structure before the annealing process. According to [32], the <111> 60° twin in the original microstructure took 13.35%. After the heat treatment process, the main annealing twin morphology was lamella-like annealing twins.

A large number of twins will appear in the face-centered-cubic structured alloys with medium or low stacking fault energy like IN718 during the solution heat treatment process. The interface energy of annealing twin is 1/10^th^ of that of HABs [33], which means excellent grain boundary stability. Thus, the driving force of forming annealing twins is the decreasing of grain boundary energy. Gleiter [34] and Mahajan [35] have done lots of research on the forming mechanism of annealing twins. The “growth accident“ mechanism has been fully accepted to explain the forming of the lamella-like straight annealing twins. Firstly, the annealing twins boundaries originates from the trigeminal grain boundary, where the interface energy is relatively high. With the grain growth, the crystal planes which are parallel to the {111} atomic planes are proned to occure the atomic misarrangement, forming stacking faults. The stacking faults enhance the system free energy. In order to reduce the free energy of the system, the subsequent order of atomic stacking on the {111} atomic plane requires stacking in a mirror-symmetrical manner with the fault to improve the symmetry of the system. That is, twin boundaries are formed during {111} atomic plane migration. Figure 11a reveals the stacking faults observed by TEM under the as-deposited condition. The stacking faults have proved the possibility of the “growth accidents” mechanism.

### 3.3. Microhardness

The microhardness of the sample in different heat treatment conditions is presented in Figure 12. The strengthening mechanisms are ascribed to the substructure strengthening and precipitate strengthening mechanism. According to the substructure strengthening mechanism, it can be apparently seen that the hardness of H930, H980, H1030 and H1080 was higher than H1130. Apparently, the recrystallization process of H1130 decreased the NF of LABs. Essentially, the LABs strengthening originated from the dislocation. The LABs with high MAs included a large number of dislocations. The NF of LABs can be used to distinguish the high dislocation density versus the low one, while the KAMA distribution indicates the degree of plastic strains, which is nearly associated with the dislocation density. Strictly speaking, the strengthening mechanism should be the strain strengthening. Figure 11b illustrates the dislocation tangles in the as-deposited condition. From 980 °C to 1080 °C, the microhardness held on 300 HV, since there is no obvious of the LABs and KAMAs distributions. After 1090 °C, with the process of static recrystallization, the microhardness decreased gradually; when heated to 1130 °C, the microhardness has dropped to 234.9 HV, 73% of the as-deposited condition, which means the residual stress was released after the recrystallization process. To be noted, under the condition of 930 °C, the microhardness increased to 331 HV, because two factors: (1) The substructure was not eliminated. The strain strengthening also existed under this condition; (2) The forming of precipitate γ″, which is coherent with the matrix. The coherent strain induced on the interface between the matrix and δphase inhibited the movement of the dislocation, which increased the hardness of the sample. However, the coarse γ″ is prone to transform into the equilibrium phase δ, which can’t be serviced under high temperature. The microhardness of the as-rolled IN718 is 160–240 HV [36], which is the same level of the SLMed IN718 in solution heat treatment condition. The strain strenghtening generated from the printing process enhanced the microharness effectively due to the fast solidifaction and melting.

## 4. Conclusions

The precipitates, grain features and hardness of IN718 alloy under the as-deposited and different solution heat treatment conditions were investigated in this article. The conclusions can be summarized as follows:
The precipitates of as-deposited IN718 are mainly Laves phases. After solution heat treatment under 930 °C/h, the coarse γ″ and δ precipitate from the matrix. When heated above 980 °C, most Laves phases dissolved into the matrix; only particle Laves phases and small bulk Laves phases can be seen along subgrain boundaries and grain boundaries, respectively. Even higher temperature can’t dissolve all the Laves phases.The aspect ratio of grains under as-deposited condition is 1.66. From 980 °C to 1080 °C, the aspect ratio stayed around 1.6, but dramatically dropped down to 1.03 when heated to 1130 °C. Static recrystallization process occurred at 1090 °C and was completed rapidly at 1130 °C.The LABs with high KAMAs kept stable during 980–1080 °C, and decreased hastily after 1090 °C. LABs disappeared completely at 1130 °C. The forming of annealing twins accompanied the recrystallization process. At the end of the recrystallization process, the NF of annealing twin boundaries reached 65%.The microhardness is affected by both the precipitates and grain features. The appearance of coarse γ″ can enhance the microhardness due to the coherent strain strengthening at 930 °C. Substructure strengthening holds on from 980–1080 °C; residual strain didn’t disappear during this temperature interval.

## Figures and Tables

**Figure 1 materials-13-00340-f001:**
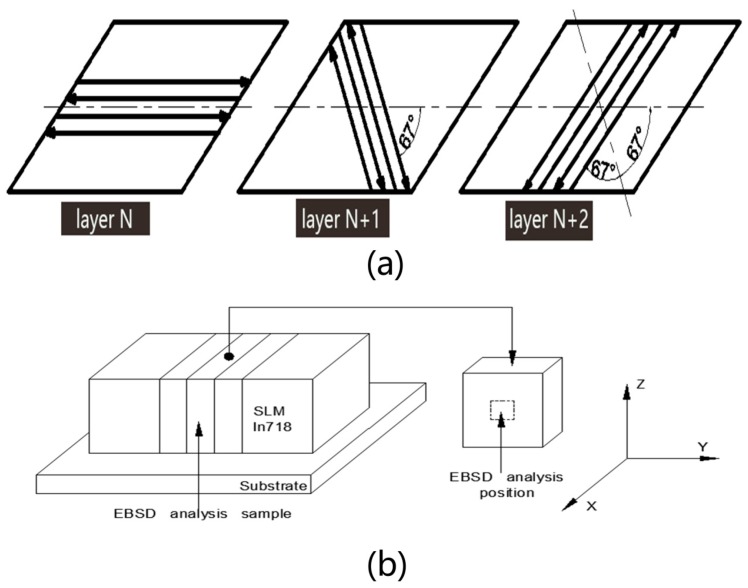
(**a**) Scanning strategy sketch map. (**b**) Sketch map of sample cutting and electron backscatter diffraction (EBSD) analysis sections (Z direction is the building direction).

**Figure 2 materials-13-00340-f002:**
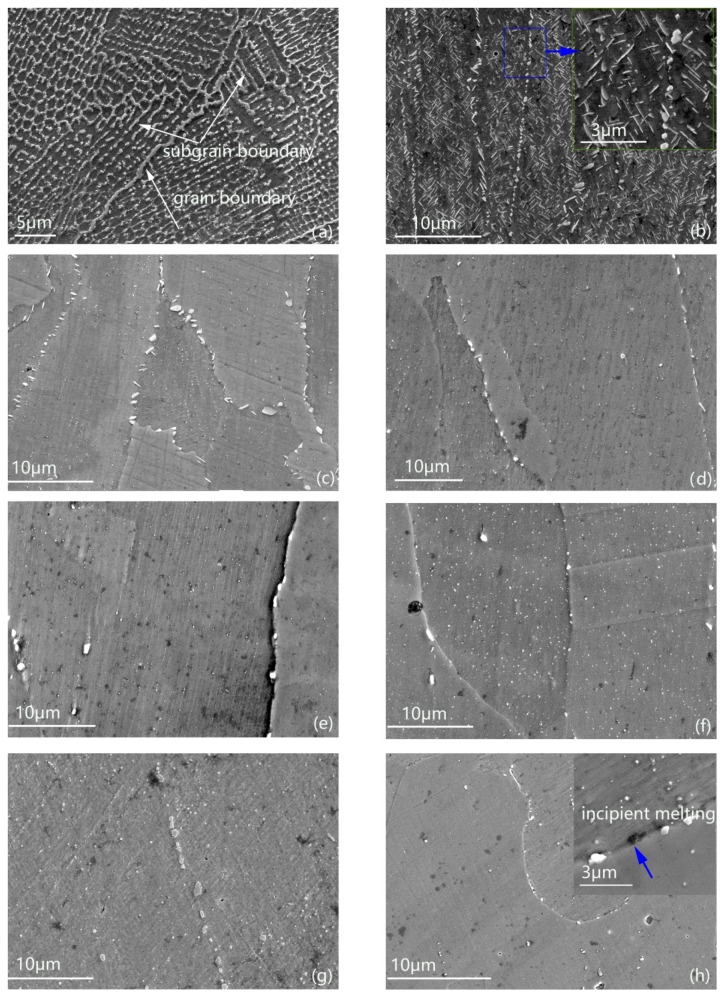
Precipitates distributions of as-deposited sample and solution-treated samples at different temperatures: (**a**) As-deposited, (**b**) 930 °C, (**c**) 980 °C, (**d**) 1030 °C, (**e**) 1080 °C, (**f**) 1130 °C, (**g**) 1180 °C, (**h**) 1230 °C.

**Figure 3 materials-13-00340-f003:**
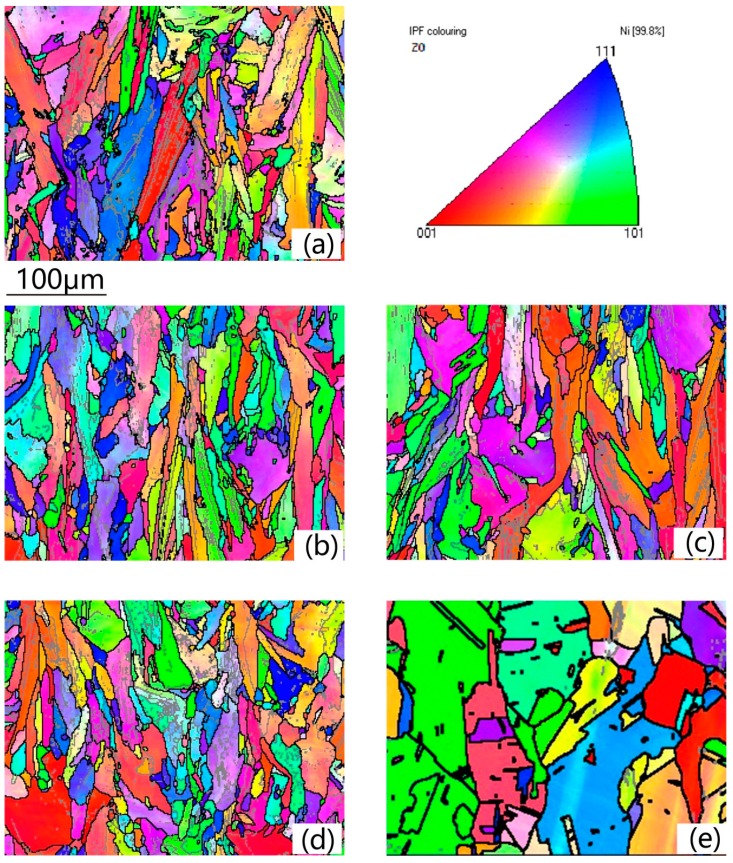
EBSD inverse-pole figure of the samples: (**a**) As-deposited condition, (**b**) 980 °C/h, (**c**) 1030 °C/h, (**d**) 1080 °C/h, (**e**) 1130 °C/h.

**Figure 4 materials-13-00340-f004:**
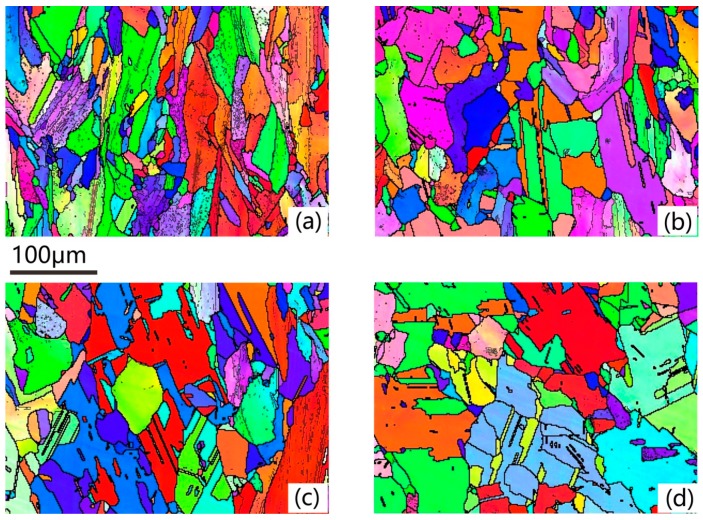
EBSD inverse-pole figure of the samples, (**a**) 1090 °C/h, (**b**) 1100 °C/h, (**c**) 1110 °C/h, (**d**) 1120 °C/h.

**Figure 5 materials-13-00340-f005:**
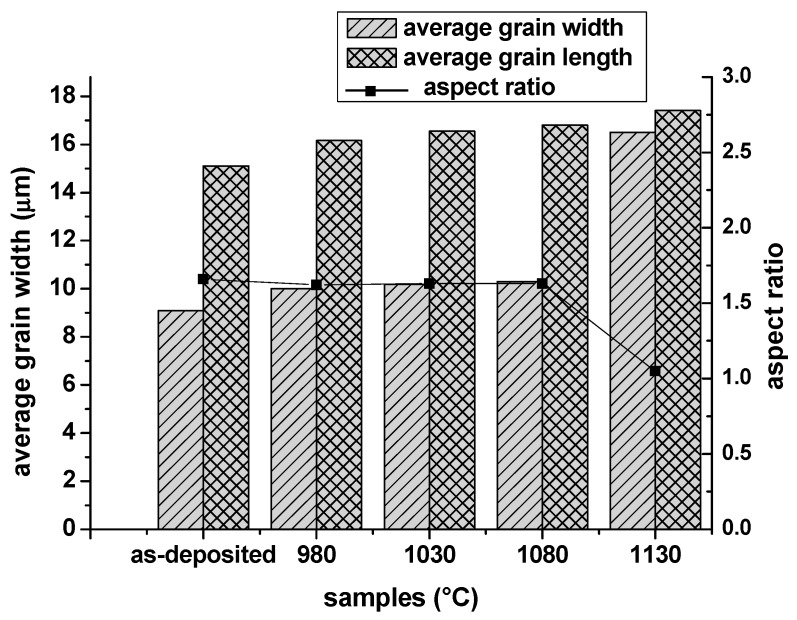
Average grain width, average grain length and grain width–length ratio of samples in different solution heat treatment conditions.

**Figure 6 materials-13-00340-f006:**
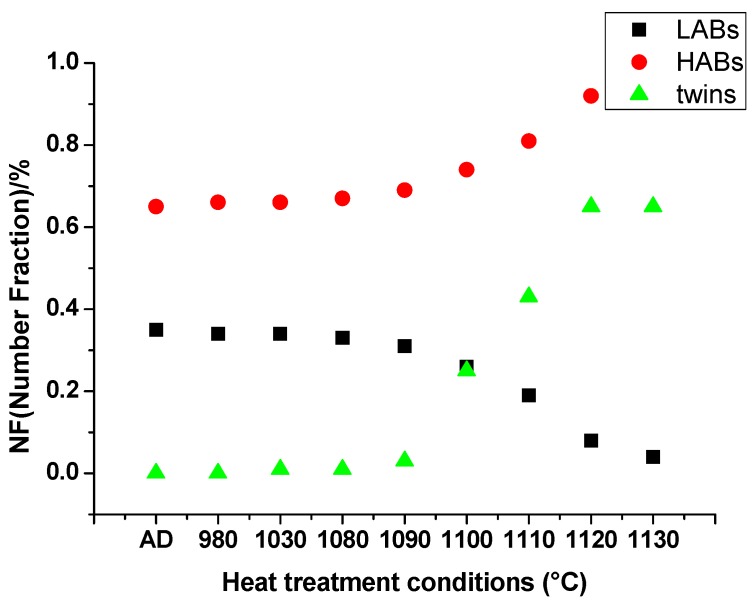
Grain boundary distributions of as-deposited condition and different solution heat treatment conditions.

**Figure 7 materials-13-00340-f007:**
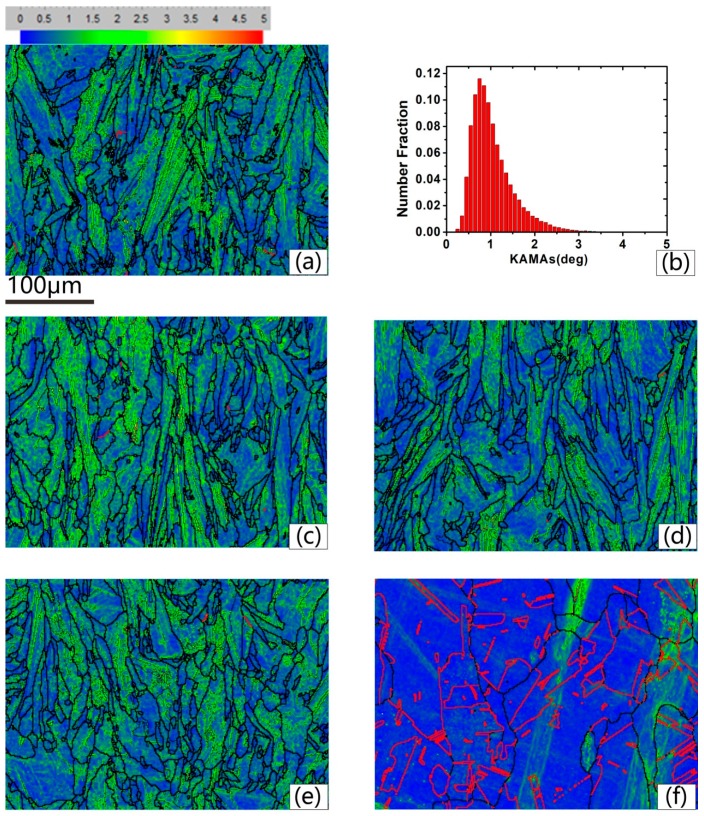
Kernel average misorientation angles (KAMAs) distribution and <111> 60° annealing twins boundaries distribution under different conditions: (**a**) As-deposited condition, (**b**) number fraction (NF) of KAMAs under as-deposited condition, (**c**) 980 °C, (**d**) 1030 °C, (**e**) 1080 °C, (**f**) 1130 °C.

**Figure 8 materials-13-00340-f008:**
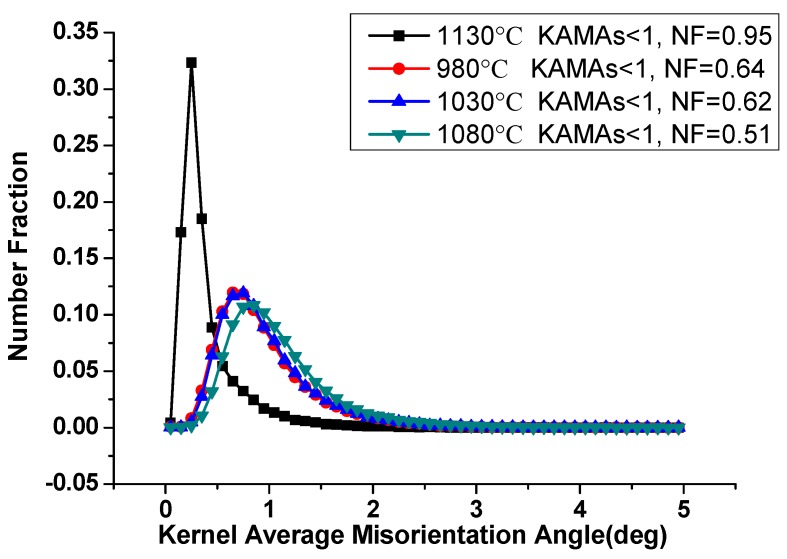
NF of KAMAs of 980 °C/1030 °C/1080 °C/1130 °C.

**Figure 9 materials-13-00340-f009:**
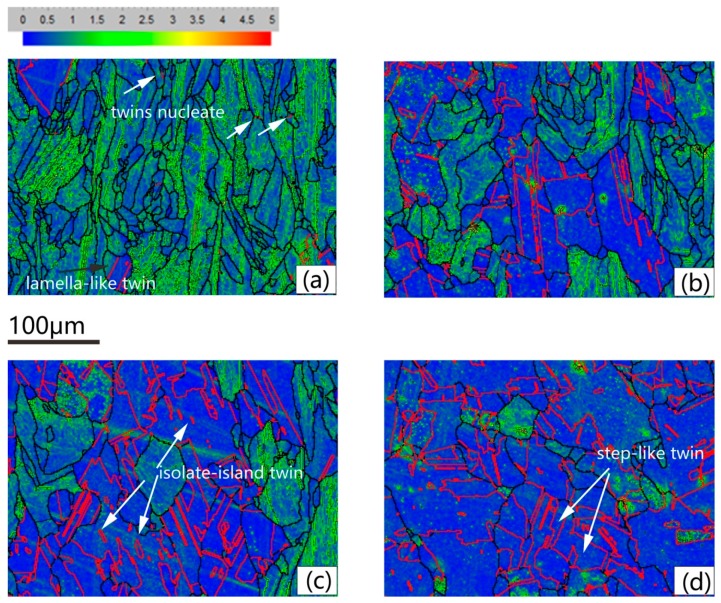
KAMAs distribution and <111> 60° annealing twins boundaries distribution under different conditions: (**a**) 1090 °C, (**b**) 1100 °C, (**c**) 1110 °C, (**d**) 1120 °C.

**Figure 10 materials-13-00340-f010:**
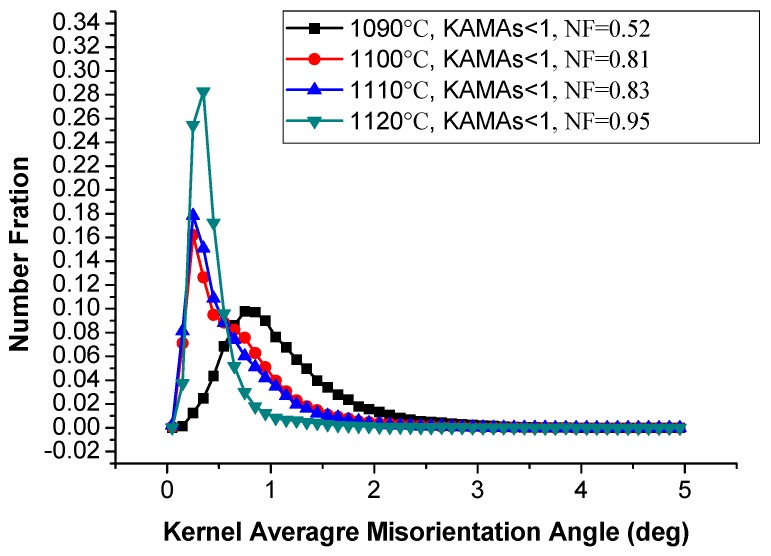
NF of KAMAs of 1090 °C/1100 °C/1110 °C/1120 °C.

**Figure 11 materials-13-00340-f011:**
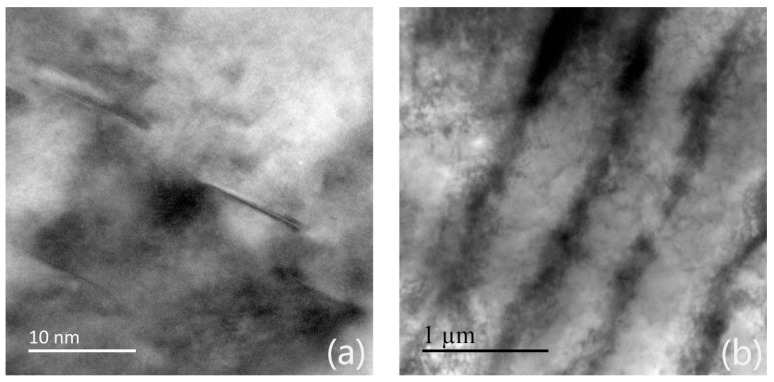
(**a**) TEM observation of the stacking faults in the as-deposited condition; (**b**) dislocations in the as-deposited condition.

**Figure 12 materials-13-00340-f012:**
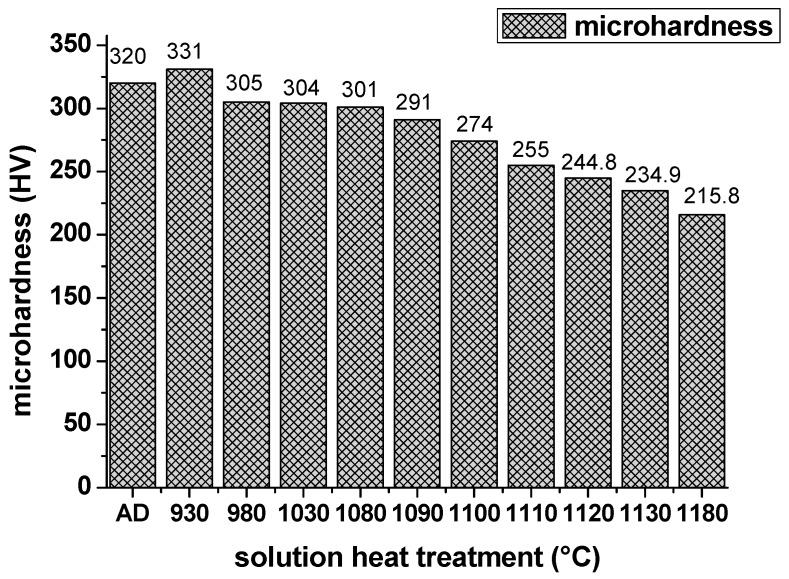
Microhardness of samples under the as-deposited condition and different solution heat treatment conditions.

**Table 1 materials-13-00340-t001:** Chemical composition of Inconel 718 (IN718) powder (mass fraction: %).

Element	Al	Ti	Cr	Fe	Nb	Mo	Ni
wt %	0.6	1.0	19.7	18.4	5.1	3.0	Bal.

**Table 2 materials-13-00340-t002:** Solution heat treatment process parameters.

Designation	Solution Treatment	Designation	Solution Treatment
As-deposited	×	H1090	1090 °C/h/WC
H930	930 °C/h/WC	H1100	1100 °C/h/WC
H980	980 °C/h/WC	H1110	1110 °C/h/WC
H1030	1030 °C/h/WC	H1120	1120 °C/h/WC
H1080	1080 °C/h/WC	H1130	1130 °C/h/WC

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
