# Peer review of "Effect of the Solution Temperature on the Precipitates and Grain Evolution of IN718 Fabricated by Laser Additive Manufacturing"

_materials, 2020, doi:10.3390/ma13020340_

Round 1
Reviewer 1 Report
This paper studied the effect of the solution heat treatment temperature on both the precipitates and grain boundary evolution of additive manufactured Inconel718, printed by selective laser melting. Recrystallization process and formation of annealing twins were examined. The effect of precipitates and grain boundaries on the hardness was evaluated. However, this paper did not state clearly what the main scientific contribution was. Besides, there are many typo all over the paper. The reviewer ask the authors to reply and modify the paper following the comments below:
-There is a typo in abstract, line 24; ‘my’ should be ‘by’.
Line 39; (.) should be (,)
Line 78; (10) should be (100)
- The introduction section does not provide sufficient background and not explain innovative part of this work.
-In section 2.3, line 72, the authors stated that “ the specimens were mechanically polished and followed by electropolishing in a solution of 10% perchloric acid ethanol at 25V for 30 seconds to produce a strain-free surface“. What is the exact composition of Perchloric Acid and Ethanol? The rate of Electropolishing is usually higher on the corners and edges compared to centers of a sample. How did you make sure the surface finish is uniform all around the sample’s surface?
-Print path needs to show in figure 1 since the mechanical and grain features depend on it.
-In figure 2, the SEM images may not appropriate. It could be some space between SEM images to become easy to read.
-In section 3.1, line 102, the authors stated that “In the Fig.2 (a), the Laves phase distributed along the grain boundaries and subgrain boundaries to form a chained-like connection, which has clearly displayed the grain boundaries and dendrite-cellular substructure inside of the grains”. However it is not obvious form the figure, the authors can insert better resolution images and make the mentioned areas clear by arrows on the SEM image.
- In section 3.1, line 106, the authors stated that “Fig.2 (b) illustrates more precipitates with profuse morphologies. The acicular precipitates are coarse γ″ and the short-rod precipitate is δ (Ni3Nb), which is in accordance to the references”. Again the authors can show the phases by arrows on the SEM image.
-In section 3.1, line 118, the authors stated that “During this temperature interval from 1030 ℃ to 1230 ℃, with the rising of the temperature, the dissolved amount of Laves phases maintained the same”. How can you confirm that it is the same? I suggest the authors should do some other characterization analysis that can show the amount of different phases quantitatively to support it.
-In section 3.1, line 121, the authors stated that “incipient melting phenomenon has been found in the 1280 ℃ solution condition, as indicated in the Fig.2 (h)”. It is 1230 ℃ not 1280 ℃ ( typo). In addition, it is not obvious from the fig 2 (h), this sentence should be revised accordingly.
-In figure 5, aspect ratio legend is missing.
-In section 3.2.1, line 159, the authors stated that “ Under the solution heat treatment condition of 1130 ℃/1h in Fig. 3 (e), the grain width becomes bigger and grain morphology has changed into equiaxed grain, the aspect ratio decreased to 1.03, which means complete recrystallization has finished” Why the recrystallization has been completed? There is no enough scientific consideration for this sentence. How can the authors confirm that at 1130℃ recrystallization has finished?
- There is a typo in figure 4 (c ); (1100℃) should be (1110℃)
-In section 3.2.1, Line 190, how big and small is defined here? Big grain size compared to what?
- In figure 5, the authors can include aspect ratios between 1080 to 1130 ℃ as figure 4 (for example for 1090, 1100, 1110, and 1120℃) to support the relevant discussion in this section.
- In section 3.2.2, Line 199, the authors stated that “Fig.3 also shows the distribution of the LABs and HABs, which are represented by the gray full lines and black full lines, respectively”. Gray and black full lines can not be realized in the figure 3, the authors can make it more clear.
-In figure 6, Standard deviations of Number Fraction are not shown in the figure and in the text .
-In figure 8, how were the number fractions calculated in the legend? They are different from the relevant numbers on the vertical axis. Should they be the same? The authors should clarify the calculation method of NF.
-There are typos in fig 9; the letters do not match; (a) 1090 ℃ (d) 1100 ℃ (e) 1110 ℃ (f) 1120 ℃ should be (a) 1090 ℃ (b) 1100 ℃ (c) 1110 ℃ (d) 1120 ℃.
-In section 3.2.3, Line 262; the authors stated that “ The isolate-island-like annealing twins indicated by white arrow in Fig.9 (c) are isolated inside of the grain”. There is not such white arrows. Show White arrows on figures 9 (c) and (d) .
-In figure 12, Standard deviations of microhardness of the samples in different heat treatment conditions are not shown. Is it enough low?
- The author should compare the results (hardness, grain features, NF, etc…) of the heat treated, 3D printed IN718 with other researchers used additive manufacturing to produce IN718 parts and/or other conventional manufacturing methods.

Reviewer 2 Report
The authors of manuscript the effects of the solution heat treatment temperature on the precipitates, grain boundary evolution and response of the microhardness of IN718 superalloy fabricated by selective laser melting (SLM) were investigated.
As the authors rightly noted the SLM is a promising additive manufacturing technique which can form metal parts with complex shape and as well as provide fast solidification rate can result much finer microstructure and uniform element distribution.
However most doubts are focused on the dissolution of detrimental precipitate generated from SLM process and the precipitation of the strengthening phase γ″ during the aging process.
Also in the literature a little attention has been focused on the grain boundary evolution during the solution heat treatment process.
The process of recrystallization of grains and change of temperature as a result of the applied treatment is clearly and understandable described in manuscript.
The presentet manuscript contains interesting results about effect of the solution temperature on the precipitates and grain evolution of IN718 fabricated by laser additive manufacturing.
The authors used the LAB can be used to distinguish between high dislocation density compared to low as well as they used the KAMA's distribution indicates the degree of plastic deformation.
The topic of the manuscript is interesting and discusses many issues. The authors clearly described the changes associated with the participation of the Laves phase.
However should be consider the minor comments:
Figure 1 is unnecessary because the methodology in the text is well explained. Microstructures which present the precipitates distributions of as-deposited sample and solution-treated samples at different temperatures in Figure 2 should be a little bigger. It would be better to present the results in two columns, then the drawings of the microstructures will be larger and more readable.
Round 2
Reviewer 1 Report
I went through the reply from the authors and found them satisfying. I believe after this revision, the paper is acceptable.